# Proinflammatory Cytokines, Type I Interferons, and Specialized Proresolving Mediators Hallmark the Influence of Vaccination and Marketing on Backgrounded Beef Cattle

**DOI:** 10.3390/vetsci12090834

**Published:** 2025-08-30

**Authors:** Hudson R. McAllister, Sarah F. Capik, Kelsey M. Harvey, Bradly I. Ramirez, Robert J. Valeris-Chacin, Amelia R. Woolums, Brandi B. Karisch, Paul S. Morley, Matthew A. Scott

**Affiliations:** 1Veterinary Education, Research, and Outreach Program, Texas A&M University, Canyon, TX 79015, USA; 2Tumbleweed Veterinary Services, PLLC, Amarillo, TX 79118, USA; 3Department of Veterinary Pathobiology, College of Veterinary Medicine, Oklahoma State University, Stillwater, OK 74078, USA; 4Prairie Research Unit, Mississippi State University, Prairie, MS 39756, USA; 5Department of Pathobiology and Population Medicine, College of Veterinary Medicine, Mississippi State University, Mississippi State, MS 39762, USA; 6Department of Animal and Dairy Sciences, Mississippi State University, Mississippi State, MS 39762, USA

**Keywords:** cattle, marketing, vaccine, RNA, bovine respiratory disease, inflammation, transcriptomics

## Abstract

This study explores how early-life vaccination and marketing decisions affect the immune and metabolic responses of beef calves. Viral vaccination programs and marketing decisions early in a calf’s life can have major implications on bovine respiratory disease (BRD) risk in subsequent beef production phases. Using a randomized controlled trial, 41 healthy calves were either vaccinated with a commercially available viral vaccine or not, then shipped directly to a backgrounding facility or through a commercial auction system. Blood samples collected at five time points were analyzed using RNA sequencing to detect changes in gene expression. The most notable immune activity occurred at the start of backgrounding, particularly in calves that were not vaccinated/marketed through a sale barn (NOVAX/AUCTION). These calves demonstrated strong activation of inflammatory and innate immune genes. In contrast, vaccinated calves that went through a sale barn (VAX/AUCTION) displayed a more moderated immune response. Calves shipped directly to backgrounding, regardless of vaccination status, demonstrated the most stable gene expression patterns over time. These findings highlight that both vaccination and marketing strategy significantly influence immune readiness and inflammatory mediation in calves.

## 1. Introduction

North American beef production systems are highly dynamic, complex systems with a multitude of factors and decision points that influence health, performance, and economic sustainability [1]. Crucially, the importance of early life management decisions on bovine respiratory disease (BRD) risk in beef cattle cannot be ignored, as the complete pathogenesis remains debated [2,3]. Moreover, management decisions such as weaning, transportation, vaccination, and marketing strategy are known to influence BRD incidence [4,5]. Specifically, certain aspects of management such as vaccination and weaning strategies have been more widely studied regarding their impact on BRD incidence, while others such as marketing strategies are less understood [6,7,8]. However, it has been demonstrated that cattle that are transported increasingly long distances or are sent through a commercial auction system (i.e., sale barn) are often immunologically stressed and at increased risk for BRD [9,10]. However, the exact distance, transport conditions, or marketing conditions that equal the most risk are unknown. Moreover, it is currently unknown how the added component of previous viral vaccination influences immunological stress when cattle undergo these marketing and transportation conditions. Variation in BRD incidence within groups exposed to the same conditions and the fact that some risk factors are often confounded with other risk factors further complicates BRD risk assessment [7,9,11].

A better grasp of the complex dynamics related to management within beef production systems is key to understanding how to better manage BRD risk [1]. It is often assumed that physiological stress can have cascading effects on the immune system and subsequent proinflammatory patterns and risk of disease development [12]. Newer tools such as advanced sequencing techniques can provide further insight into the effects of management factors and common health interventions on host gene expression, immune function, health, and performance. However, remaining knowledge gaps limit our ability to appropriately mitigate or predict BRD risk and reduce the productivity and sustainability of beef production.

As such, our objective was to evaluate the interaction of marketing strategy and attenuated viral vaccination for respiratory pathogens on host gene expression in young beef calves that remained clinically healthy from birth through their backgrounding phase of life. This study supports the exploration of associations regarding the impact of two common management events, preweaning vaccination and marketing, that occur in the beef production supply chain. In addition to exploring associations, this study generates novel hypotheses to improve the understanding of how marketing and preweaning vaccination influence long-term health outcomes. We hypothesize that commonly used management tactics, namely preweaning vaccination and marketing strategy, influence gene expression patterns associated with immune modulation and inflammatory mediation.

## 2. Materials and Methods

### 2.1. Animal Use and Study Enrollment

This study was carried out in accordance with Animal Research: Reporting of In Vivo Experiments (ARRIVE) guidelines [13]. Forty-one calves that remained clinically unaffected by BRD throughout the cow-calf and backgrounding phases of production were selected via stratified random sampling from a study population previously described [14,15]. We elected to use this number of cattle based on an *a priori* power calculation. Our prior data indicated that the minimum average read counts among the prognostic genes in a group was 200 counts, the maximum dispersion between samples was typically 0.4, and the ratio of the geometric mean of normalization factors was approximately 1.2 [14,15]. Assuming that there will be approximately 16,000 filtered genes for differential expression analysis and the top 200 genes are prognostic with minimum fold change between groups set to 2 and utilizing a false discovery rate (FDR; adjusted *p*-value) cutoff of 0.05, this experiment was designed to provide 82% statistical power to detect differentially expressed genes between cattle [16].

Briefly, bull calves were enrolled in a split-plot randomized controlled trial to evaluate the impact of different management strategies on BRD morbidity, mortality, and performance [15]. Cattle were randomly assigned to the whole-plot level treatment, which involved vaccination with 2 mL Pyramid 5 (Boehringer Ingelheim Animal Health, Duluth, GA, USA) subcutaneously (VAX) or being given 2 mL 0.9% saline subcutaneously (NOVAX) on T1 (median age = 107 days) and T3 (median age = 183 days). Seven days post-vaccination (T2; median age = 114 days), all calves were weighed and sampled. All calves received a multivalent clostridial bacterin-toxoid vaccine subcutaneously (Covexin 8; Merck Animal Health, Rahway, NJ, USA) on T1 and were given a booster with the same vaccine and surgically castrated on T3. Calves were abruptly weaned at T4 (median age = 230 days) and randomly assigned to split-plot level treatment, which involved being weaned in their original pastures in Mississippi (Prairie Research Unit, Prairie, MS, USA) for three days prior to direct transportation from Mississippi to Texas for backgrounding (DIRECT) via commercial livestock trailering or sent to a commercial auction market where they were housed in a holding pen not in direct contact with other cattle for approximately six hours, then transported for housing at an order buyer facility for three days prior to transport to Texas (AUCTION). Cattle not used in this study (i.e., those housed at the order buyer facility at the same time as the study cattle) were from other commercial sources in the surrounding area but were neither mixed nor in direct contact with the study cattle. Notably, all cattle were transported to the background facility in Texas simultaneously in a single commercial livestock trailer from Mississippi to Texas (approximate 13 h transit; Texas A&M AgriLife Bushland Research Feedyard, Bushland, TX, USA). Here, DIRECT and AUCTION cattle were separated by trailer level to prevent direct contact between the two groups; DIRECT calves were loaded first onto the second floor, followed by the AUCTION cattle on the first floor. Upon arrival in Texas (T5; four days post-weaning), calves were kept in one of 12 pens corresponding to their original random assignment to whole and split-plot treatments. All cattle were monitored daily for signs of clinical BRD throughout the cow-calf and backgrounding periods by trained observers who were blinded to treatment (vaccination, marketing strategy); the clinical scoring system used in this study for BRD diagnosis has been previously described [15]. Briefly, cattle were assigned a clinical BRD score of 0–4 based on visual signs of disease, where detailed health histories were kept every day for each calf throughout the cow-calf and backgrounding phases. While there were no further laboratory-based analyses conducted to confirm BRD diagnosis in this study, this reflects common industry practices. The five sampling timepoints in this study were based on common beef industry management practices of initial respiratory vaccination or branding vaccination, revaccination, and weaning, while also capturing critical biological instances of immunological development. Whole blood was collected from all calves into Tempus RNA blood tubes (Applied Biosystems, Waltham, MA, USA) via jugular venipuncture at each time point. The 41 cattle that were selected in this study (n = 12 VAX/DIRECT, n = 7 NOVAX/DIRECT, n = 11 VAX/AUCTION, and n = 11 NOVAX/AUCTION) resulted in 201 blood samples across the five time points analyzed for whole blood transcriptomics; four samples were either missing or failed RNA extractions. Complete metadata on all samples for these 41 cattle are provided in Appendix A.

### 2.2. Next-Generation RNA Sequencing and Bioinformatic Data Processing

Total RNA was isolated with a Tempus Spin RNA Isolation Kit (Applied Biosystems, Waltham, MA, USA), based on the manufacturer’s instructions. Total RNA from each sample was analyzed for RNA concentration and integrity with a Qubit 4.0 Fluorometer (ThermoFisher, Waltham, MA, USA) and an Agilent 4200 Bioanalyzer (Agilent, Santa Clara, CA, USA), respectively. RNA samples were considered to possess high quality (RIN: 7.6–9.7; mean = 8.9, s.d. = 0.4) and concentrations (ng/μL: 37.1–874.4; mean = 207.7, s.d. = 94.9); four samples (J009 at T5 (VAX/DIRECT); J018 at T3 (NOVAX/DIRECT); J025 at T3 (VAX/AUCTION); J117 at T3 (NOVAX/AUCTION)) failed initial RNA extraction and were removed from the study. Library preparation for mRNA was performed with the Stranded mRNA Library Prep Kit (Illumina, San Diego, CA, USA) following the manufacturer’s instructions. Paired-end sequencing for 150 base pair read fragments was performed on an Illumina NovaSeq 6000 analyzer (v1.7+; S4 reagent kit v1.5) across five flow cell lanes; samples were randomly allocated to each flow cell lane to reduce sequencing lane effects. Notably, a portion of the sequencing data utilized in this study are from our previous work [14,15].

Following RNA sequencing and read demultiplexing, quality assessment of reads was performed with FastQC v0.11.9 (https://github.com/s-andrews/FastQC; accessed 2 August 2024) and MultiQC v1.12 [17], and read trimming was performed with Trimmomatic v0.39 [18]. Trimmed reads were mapped and indexed to the bovine reference genome assembly ARS-UCD2.0 with HISAT2 v2.2.1 [19]. Sequence Alignment/Map (SAM) files were converted to Binary Alignment Map (BAM) files, prior to transcript assembly, with Samtools v1.14 [20]. Transcript assembly and gene-level expression estimation for differential expression analysis were performed with StringTie v2.1.7 [21], as described by Pertea and colleagues and our previous studies [14,15,22]. Raw sequencing data previously generated and incorporated into this study are discussed elsewhere and at the National Center for Biotechnology Information Gene Expression Omnibus (NCBI-GEO) under the accession numbers GSE205004 and GSE218061 [14,15]. New raw RNA sequencing data produced by this study are publicly available at the NCBI-GEO under the accession number GSE248477.

### 2.3. Differential Expression and Gene Trajectory Analyses

Gene-level count matrices were processed and analyzed with R v4.1.2 via the RStudio integrated development environment. Samples were first classified by vaccination group, sale type, and time point. Following sample classification, the ComBat_seq function in the package sva v3.52.0 was utilized for batch effect adjustment, in the form of sequencing runs (i.e., data from GSE205004 and GSE218061 as “1” and new data generated under GSE248477 as “2” (“Batch”; Appendix A)), using an empirical Bayes framework in the raw gene counts [23,24]; this was applied to all sequencing libraries simultaneously. Adjusted gene counts were then processed and filtered by procedures described by Chen and colleagues [25]; genes with a minimum total count above 100 and a count-per-million (CPM) of 0.2 in at least 24 samples were retained.

For pairwise testing, gene counts were normalized for differential expression analysis with the trimmed mean of M-values method [26]. Following common and tagwise dispersion estimation, negative binomial pairwise comparisons for DEGs between T4 and T5 within each treatment group were performed with edgeR v3.36.0 [25,27], fitting genes under a generalized linear model framework and employing quasi-likelihood F-tests [25]. Specifically, we employed a reduced model to account for housing (pasture in Mississippi) and individual ID (paired sampling) when appropriate, where any gene was considered differentially expressed with an FDR < 0.05. Here, we focused on the differences across T4 and T5 within each treatment group (NOVAX/AUCTION, VAX/DIRECT, etc.) and differences across treatment groups at T5 to explore the interaction of treatments and host gene expression at the time of backgrounding (i.e., the collective impact of the split-plot design). Differentially expressed genes were subsequently analyzed for functional enrichment of gene ontology (GO) terms, Reactome pathways, and KEGG pathways with KOBAS-i [28] (accessed 23 October 2024), utilizing hypergeometric testing against the bovine (*Bos taurus*) reference species and Benjamini-Hockberg adjusted *p*-values (FDR) < 0.05. An UpSet plot was generated to visualize the overlap of DEGs identified between T4 and T5 within each split-plot group and between intervention groups at T5 via Intervene [29,30]. Heatmap visualization and clustering of gene expression at T4 and T5 was performed with the Bioconductor tool pheatmap v1.0.12 (https://CRAN.R-project.org/package=pheatmap; accessed 20 November 2024). Specifically, log2-count-per-million-transformed values were utilized from all filtered and normalized gene counts (n = 17,984), with grouping of gene expression patterns into 48 distinct clusters with the k-means algorithm embedded within pheatmap based on heuristic cluster number determination via the Elbow method. Ward’s method of unsupervised hierarchical clustering on Manhattan distances and Pearson correlation coefficients were utilized across samples and gene patterns, respectively. Color scaling was performed with the Bioconductor package viridis v0.6.2 (https://cran.r-project.org/web/packages/viridis/vignettes/intro-to-viridis.html; accessed 20 November 2024) to allow ease of visual interpretation for individuals with color blindness.

To evaluate expressional trajectories of genes within each split-plot-level treatment group (NOVAX/AUCTION, VAX/DIRECT, etc.), segmented regression analyses were performed within each group across T1-T5 via Trendy v1.26.0 [31]. Filtered gene counts were normalized with the Median Normalization technique [32] within EBSeq v1.9.1 [33]. Following gene count normalization, model initialization for all four treatment groups across all five time points was performed per Trendy authors’ recommendation, utilizing a maximum number of breakpoints to consider (maxK) of 1, a minimum number of samples required to be within a segment (minNumInSeg) of 3, and the number of different seeds to try (numTry) set at 10; all other parameters were set to default. All genes where segmented regression analysis converged and identified a breakpoint or one expressional trend were retained. Genes were retained having a regression fit adjusted R2 value of > 0.50, where any gene that possessed a segment slope having an adjusted *p*-value of < 0.10 was then considered dynamically expressed.

## 3. Results

Read mapping and alignment of the 201 transcriptomes to the ARS-UCD2.0 bovine reference genome resulted in an average mapping rate of 95.50% (s.d. = 0.96%). In total, gene-level alignment resulted in a total of 32,297 unique features, with a median library size of 40,974,411 (range = 12,368,112–64,917,549). Pre-processing and filtering of low expression values resulted in a total of 17,984 genes used for downstream analyses.

### 3.1. Differential Gene Expressions and Functional Enrichments

Analysis of genes via edgeR resulted in 1037, 4234, 1432, and 205 DEGs when evaluating the VAX/AUCTION, NOVAX/AUCTION, VAX/DIRECT, and NOVAX/DIRECT, respectively; comparative analyses for DEGs via edgeR between vaccination status within each marketing type resulted in 66 and 12 DEGs between AUCTION and DIRECT cattle, respectively (Appendix A). Visualization of the number of DEGs identified from differential expression testing, along with overlapping between analysis comparisons, is found in Figure 1. Here, the DEGs identified between T4 and T5 within the NOVAX/AUCTION group were the most distinct, comprising 2809 uniquely identified genes, followed by the VAX/DIRECT and VAX/AUCTION groups with 519 and 177 uniquely identified DEGs, respectively. The highest number of shared DEGs was found between the VAX/AUCTION and NOVAX/AUCTION comparisons with 553 DEGs. Heatmap and unsupervised clustering analysis was performed to determine global gene expression patterns (Figure 2). Unsupervised clustering demonstrated that regardless of vaccination status, time was the most influential driver of global gene expression patterns, with marked clustering of marketing strategy at T5 with particular emphasis of the AUCTION group.

Functional enrichment analyses of DEGs identified with edgeR are reported by each treatment group (Appendix A). In the VAX/DIRECT comparison from T4 to T5, 118 pathways and 164 GO terms were identified. These enrichments primarily involved the immune system (innate and adaptive immunity; cytokine signaling; neutrophil degranulation; interleukin and TNF pathways; Th17 cell differentiation; class I MHC antigen processing), metabolism and cellular biosynthesis (lipid, steroid, cholesterol, carbohydrate, and protein metabolism; insulin resistance and signaling; AMPK and adipocytokine signaling), and signal transduction pathways (MAPK, PI3K-Akt, NF-κB, mTOR, FoxO, WNT, and GPCR signaling; receptor-mediated pathways (e.g., nuclear hormone receptors, relaxin, growth hormone, cytokines)).

In the NOVAX/DIRECT comparison from T4 to T5, 29 pathways and 52 GO terms were identified. These enrichments primarily involved lipid and cholesterol metabolism (cholesterol homeostasis; cholesterol import/export; cholesterol efflux; reverse cholesterol transport; sterol biosynthetic and metabolic processes; plasma lipoprotein clearance and assembly/remodeling; apolipoprotein binding; SREBP and PPAR signaling), inflammation (positive regulation of Th1 and type I interferon mediated signaling; chemokine receptor signaling; response to bacterium and virus; defense against Gram-negative bacteria), signal transduction and regulation (MAPKKK activity; AMPK signaling; PPAR signaling; cytokine-mediated signaling; cytokine-cytokine receptor interaction; phospholipase binding; intracellular signaling cascades), and autophagy and digestion (lysosome; autophagy; fat digestion and absorption; HS-GAG degradation; intestinal absorption; phase I-functionalization of compounds).

In the VAX/AUCTION comparison from T4 to T5, 58 pathways and 70 GO terms were identified. These enrichments primarily involved lipid and steroid metabolism (cholesterol biosynthesis and homeostasis; synthesis of 5-eicosatetraenoic acids; steroid biosynthesis; fatty acid metabolism), immunity and antiviral defense (response to interferon-alpha; ISG15 antiviral mechanism; class I MHC antigen presentation; defense response to virus; innate and adaptive immune response; complement and coagulation cascades), protein modification and post-translational regulation (ubiquitin-protein transferase activity; GTP hydrolysis and ribosome joining; SUMO E3 ligases; protein ubiquitination; protein mono-ADP-ribosylation; cytoplasmic translation), and cellular stress responses (mTOR, AMPK, PPAR, and cytokine signaling; necroptosis; cellular response to DNA damage stimulus; adipocytokine signaling pathway).

In the NOVAX/AUCTION comparison from T4 to T5, 179 pathways and 213 GO terms were identified. These enrichments primarily involved immune and inflammatory signaling activation (cytokine signaling; DDX58/IFIH1-mediated induction of interferon-alpha/beta; positive regulation of interleukin-6 production; MyD88 dependent cascade initiated on endosome; neutrophil degranulation; NOD-like receptor signaling pathway; toll-like receptor cascades (e.g., TLR4, TLR7/8/9); synthesis of leukotrienes (LT) and eoxins (EX)), signal transduction and stress response (MAPK signaling pathway; mTOR signaling pathway; AMPK signaling; TNF signaling; p53 signaling pathway; signaling by NOTCH), cell cycle and DNA repair (G1/S and G2/M transition; mitotic prometaphase and anaphase; cyclin A/B/Cdk events; base excision repair; homology directed repair; global genome nucleotide excision repair (GG-NER)), and RNA metabolism (metabolism of RNA; rRNA processing in the nucleus and cytosol; mRNA surveillance pathway; nonsense-mediated decay (NMD) branches (EJC-dependent and -independent)).

When comparing the VAX/DIRECT and NOVAX/DIRECT groups at T5, 9 pathways and 47 GO terms were identified. These enrichments primarily involved lipid and cholesterol metabolism (intestinal cholesterol absorption; cholesterol import; low- and high-density lipoprotein particle clearance; cholesterol homeostasis; cortisol synthesis and secretion; aldosterone synthesis and secretion), neuronal and axonal metabolic processes (amyloid-beta clearance; long-term memory; myelination; axon fasciculation; synapse assembly; neuronal cell body membrane; somatodendritic compartment), and cellular signaling and transportation (endocytosis; phagocytosis; clathrin-coated pit; multivesicular body; early/late endosome; lysosome; potassium ion transmembrane transport; voltage-gated potassium channel complexes; potassium channel regulators). When comparing the VAX/AUCTION and NOVAX/AUCTION groups at T5, six pathways and four GO terms were identified. These enrichments primarily involved mitochondrial metabolism (oxidative phosphorylation; metabolic pathways; thermogenesis) and cell cycle regulation (APC/C:Cdh1-mediated degradation; regulation of cytokinesis; signal transduction involved in G2 DNA damage checkpoint; translesion synthesis).

### 3.2. Trend-Wise and Breakpoint Gene Expression Analyses

Dynamic trend-wise gene expression was identified for NOVAX/AUCTION (n = 1552), VAX/AUCTION (n = 3600), NOVAX/DIRECT (n = 2037), and VAX/DIRECT (n = 2879) groups across T1–T5 (Appendix A). Of particular interest, genes were considered dynamically expressed with at least one statistically significant segment slope (adjusted *p*-value < 0.10) for the NOVAX/AUCTION (n = 764), VAX/AUCTION (n = 2513), NOVAX/DIRECT (n = 625), and VAX/DIRECT (n = 1415) groups. Notably, several immune-related genes were identified that corresponded to significant functional enrichments in the previous analyses, specifically involving inflammatory resolution responses and lipid metabolism, cytokine production, and antiviral activity, such as *ALOX15*, *ALOX5*, *CATHL1*, *CD14*/*68*/*163*, *CFB*, *COL1A1*/*2*, *GPR4*, *GPX4*, *HERC4*/*5*/*6*, *HPGD*, *IFI6*, *IFIH1*, *IFIT2*/*3*/*5*, *IFITM2*/*3*/*5*, *IL1B*/*1R2*/*1RAP*/*1RN*/*12B*/*13RA1*/*16*/*17D*/*18RAP*/*21R*, *ISG15*, *LIF*, *LRP1*, *MX2*, *S100A9*/*12*, *TNF*, and *TRIM25*.

When comparing relative gene expression trends of type I interferon-stimulated genes (*IFIT2*, *IFIT3*, *IFIT5*, *IFITM2*, *MX2*, and *TRIM25*), NOVAX/AUCTION cattle (Figure 3) possessed a relative increase and earlier breakpoint (around T4) of all six of these genes when compared to VAX/AUCTION cattle (Figure 4), for which four of the six genes did not possess significant change over time (*IFIT2*, *IFIT3*, *IFITM2*, and *MX2*). Evaluation of these six genes in NOVAX/DIRECT (Figure 5) and VAX/DIRECT (Figure 6) yielded no breakpoints nor significant changes in these gene expression patterns over T1-T5.

To investigate the temporal regulation of genes associated with specialized pro-resolving mediators (SPMs) and lipid metabolism, we assessed the expression dynamics of *ALOX5*, *ALOX15*, *GPX4*, and *HPGD* across all four treatment groups (Figure 7). Time-course analysis revealed distinct patterns of gene regulation influenced by both vaccination status and marketing strategy. *ALOX5* and *ALOX15* demonstrated consistent upregulation across all four groups over the first four timepoints (i.e., cow-calf phase prior to marketing). However, the NOVAX/AUCTION group possessed a marked downregulation of these genes with a significant breakpoint between T4 and T5, while the VAX/AUCTION group maintained no significant change in gene expression trends following marketing. The DIRECT cattle, regardless of vaccination status, maintained relatively stable expression levels of these two enzymes. Expression levels of *GPX4* demonstrated significant downregulation under NOVAX/AUCTION and VAX/AUCTION conditions, while the DIRECT groups exhibited relatively stable expression over time. *HPGD* expression displayed more variable trajectories across the four groups: no relative change in expression over time followed by downregulation (breakpoint around T4) in NOVAX/AUCTION, increased expression over time with no significant change by T5 in VAX/AUCTION, and largely stable expression in the DIRECT groups, with consistently increased expression over time in VAX/DIRECT cattle. Overlapping the results from pairwise testing (edgeR; Appendix A) revealed similar trends. Here, *ALOX15*, *ALOX5*, and *HPGD* were significantly downregulated at T5 compared to T4 in VAX/AUCTION and NOVAX/AUCTION, with no significant differences within the DIRECT groups nor when comparing VAX/AUCTION to NOVAX/AUCTION at T5; *GPX4* was only significantly downregulated within the NOVAX/AUCTION cattle at T5 in comparison to T4.

## 4. Discussion

Bovine respiratory disease remains one of the most persistent health challenges in the U.S. beef industry, primarily due to its complex etiology and the decentralized, non-integrated nature of beef production systems. Key risk factors, including weaning, transportation, commingling, and castration, are well-recognized contributors to BRD [7,34]; however, the interactive effects of specific management decisions, such as preweaning vaccination and marketing decisions, are still poorly understood and inconsistently reported across studies [7,35,36].

This study represents a step toward understanding how routine management strategies shape gene expression trajectories in young beef calves. Importantly, we assessed clinically healthy animals to avoid the effects of overt disease. Even in the absence of clinical BRD, our results show differences in transcriptomic profiles influenced by both marketing strategy and vaccination history. These findings reinforce the need for future research to move beyond clinical endpoints and toward predictive, mechanistic markers of resilience and immunocompetence especially given that visual detection of BRD often lacks sensitivity and fails to identify subclinical disease [37,38].

### 4.1. Marketing Strategy Is a Driver of Gene Expression at Backgrounding Entry

The most profound transcriptional changes were observed in the NOVAX/AUCTION group, particularly at the transition from T4 to T5, coinciding with the marketing event. These calves exhibited 2809 uniquely DEGs, accompanied by enrichment of immune, inflammatory, metabolic, and stress-related pathways. This group also demonstrated abrupt breakpoints in interferon stimulating genes (ISGs), such as *IFIT2*, *IFIT3*, *IFIT5*, *MX2*, and *TRIM25*, suggesting activation of antiviral and innate immune responses that hallmark systemic inflammation, cellular stress, and, likely, viral pathogen exposure [39,40,41].

All cattle in this study experienced the stress associated with abrupt weaning and long-distance travel from Mississippi to the Texas Panhandle. In contrast to AUCTION calves, calves marketed through a DIRECT route demonstrated a more moderated transcriptional response, with significantly fewer DEGs and enriched pathways. These cattle exhibited greater transcriptomic stability over time, which may indicate a reduced exposure to commingling, novel pathogens, and transport-related stressors; these findings are consistent with previous work demonstrating lower BRD incidence and improved performance in directly shipped calves [34,42]. This contrast in transcriptomic profiles may support direct-to-backgrounding marketing as a beneficial strategy for minimizing immunological disruption, particularly in the critical transition period immediately post-weaning.

### 4.2. Vaccination Limits Immune Dysregulation in Auction Marketed Cattle

Preweaning vaccination demonstrated a marked effect on the transcriptional response to marketing, particularly under AUCTION conditions. VAX/AUCTION calves exhibited a more tapered response in both ISG expression and inflammatory mediator profiles compared to NOVAX/AUCTION cattle. Specifically, the expression of genes associated with lipid-derived SPMs (*ALOX5*, *ALOX15*, *GPX4*, and *HPGD*) remained relatively stable or were upregulated in VAX/AUCTION calves, while NOVAX/AUCTION cattle demonstrated significant downregulation by backgrounding entry (T5), likely reflecting impaired resolution of inflammation. Moreover, the VAX/AUCTION group exhibited a decrease in the expression and number of genes related to ISGs when compared to the NOVAX/AUCTION group. This suggests that preweaning vaccination can influence these innate immune and inflammatory responses driven by marketing exposure, even weeks after the vaccine is administered. As such, the immunological influence of vaccination warrants further investigation for its effect on innate immunity and inflammation in auction-marketed calves. Furthermore, these findings suggest that vaccination may prime the immune system toward controlled reactivity and promote immunological resolution rather than hyperactivation during stress or potential viral exposure/challenge [43,44,45]. Notably, this effect was not as evident in DIRECT cattle, likely because marketing-induced stressors were not sufficient to induce as strong of an immune or inflammatory response compared to AUCTION cattle.

While we did not measure pathogen burden directly, the expression of classical viral defense genes in the AUCTION cattle points to possible wild-type viral challenge in the marketing environment. The moderated expression of these genes in VAX/AUCTION calves suggests functional immune memory and supports previous findings that preweaning vaccination may reduce BRD risk in high-stress situations [45,46,47].

Interestingly, we did not observe a significant type I interferon response immediately post-vaccination, despite previous research indicating that viral vaccines and adjuvants can robustly stimulate IFN-α/β production [48,49]. One possible explanation is the sampling interval: our second time point (T2) may have occurred too long after initial antigen exposure to capture transient IFN expression. Alternatively, cattle may mount a more tissue-localized or cell-type specific interferon response not detectable via whole blood transcriptomics. Additional studies incorporating earlier sampling points and immune cell subset analysis are needed to resolve this question.

### 4.3. SPM-Related Genes Reflect Inflammatory Resolution Capacity

Expression profiles of the key SPM biosynthetic genes *ALOX5*, *ALOX15*, *GPX4*, and *HPGD* revealed an overall pattern of upregulation across the preweaning phase (T1 to T4), followed by downregulation post-marketing (T5) in AUCTION calves. In contrast, these genes remained relatively stable or progressively increased in DIRECT cattle, regardless of vaccination status. Specialized proresolving mediators are critical lipid mediators involved in resolving inflammation and restoring tissue homeostasis [50,51,52]. Their decline by T5 in AUCTION calves, particularly with NOVAX cattle, may reflect impaired resolution, potentially increasing vulnerability to poorly controlled or prolonged inflammation. Furthermore, recent studies indicate that SPMs are involved not only in resolution but also in adaptive immune regulation, modulating T-helper differentiation, promoting regulatory T cell responses, and enhancing dendritic cell function [53,54]. These immunomodulatory roles are consistent with observed enrichments in *CD28* and *CTLA4* signaling and cytokine receptors like *IL5RA* and *IL17REL*, suggesting a broader role of SPMs in coordinating long-term immunological resilience and the potential influence attenuated viral vaccination has in modulating these responses long term.

### 4.4. Study Limitations and Implications for Cattle Health and Production Systems

While this study is one of the first to evaluate specific molecular features of the bovine immune system influenced by marketing and preweaning vaccination strategies, key limitations should be considered. Although our *a priori* power calculation dictates that this study possessed adequate power for differential gene expression analyses, uneven group distribution, the use of male-only beef cattle, and the evaluation of cattle from only one year/geographical context may limit the external validity of these results and necessitates future replication studies. Only male calves were included in this study to standardize castration status, reduce sex-related variability in stress and immune responses, and reflect industry demographics. Further, all selected calves remained clinically unaffected by BRD, which, while controlling for disease status, restricts this study’s biological relevance to BRD pathogenesis and limits generalizability to other high-risk populations. Additionally, blood was evaluated at only five time points, which may not fully capture dynamic gene expression changes influenced by these management tactics. Moreover, whole blood is heterogeneous tissue and may not reflect more nuanced responses from immunocompetent tissues such as lung, spleen, or lymph nodes. Further research that emphasizes the use of more objective modalities for clinical disease and etiology detection is highly warranted.

Collectively, our findings highlight critical interactions between preweaning vaccination, marketing strategy, and temporal gene expression in young beef cattle. Based on transcriptomic evaluation, preweaning vaccination modulates immune activation during auction marketing, while direct marketing minimized inflammatory and metabolic dysregulation. These findings potentially promote industry recommendations advocating for preconditioning programs that include preweaning vaccination and direct marketing when feasible, as these practices may be associated with more stable immunometabolic profiles [7,47]. Moreover, the transcriptional changes related to lipid metabolism and immune resolution disruption in NOVAX/AUCTION cattle may indicate performance and health disadvantages in calves sold through commercial sale barns without preweaning vaccination. As such, research evaluating targeted nutritional or therapeutic interventions aimed at enhancing SPM pathways, redox balance, and stress resilience during the transition to feedlot environments is highly warranted.

## Figures and Tables

**Figure 1 vetsci-12-00834-f001:**
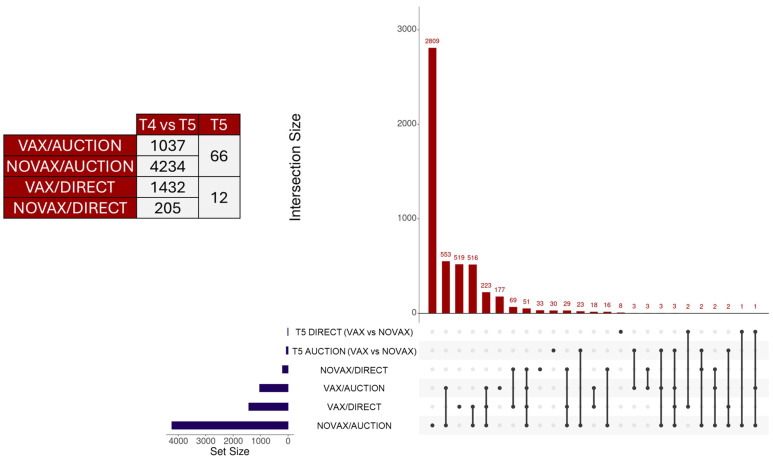
UpSet plot and differential gene expression summary across experimental comparisons of vaccination (VAX) versus sham controls (NOVAX) with respect to marketing strategy (AUCTION and DIRECT). Horizontal bars on the left represent the total number of genes identified in each individual set. Vertical bars on the top of the matrix indicate the size of each intersection between selected sets, with connected dots below each bar showing which sets are involved in that intersection.

**Figure 2 vetsci-12-00834-f002:**
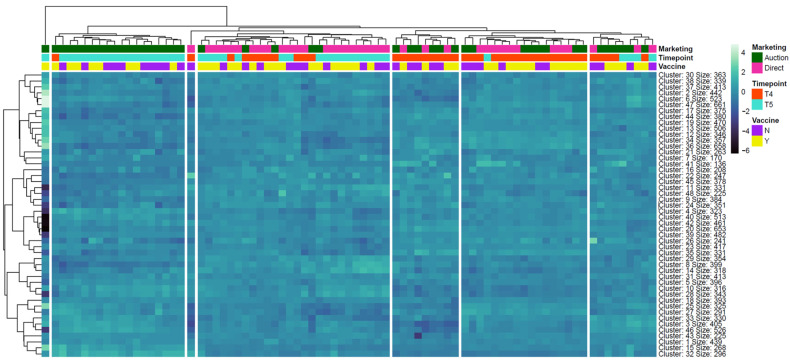
Hierarchical clustering heatmap of gene expression across timepoints, marketing, and vaccine statuses. Gene clusters (rows) and samples (columns) were hierarchically clustered based on z-score normalized gene expression values. Color annotations above the heatmap denote metadata categories: marketing (AUCTION: green, DIRECT: pink), time point (T4: orange, T5: turquoise), and vaccine status (VAX: yellow, NOVAX: purple). Expression intensity is color-coded (white/light blue = high, black/dark blue = low). Clusters are annotated with size labels indicating the number of genes per k-means cluster (n = 48).

**Figure 3 vetsci-12-00834-f003:**
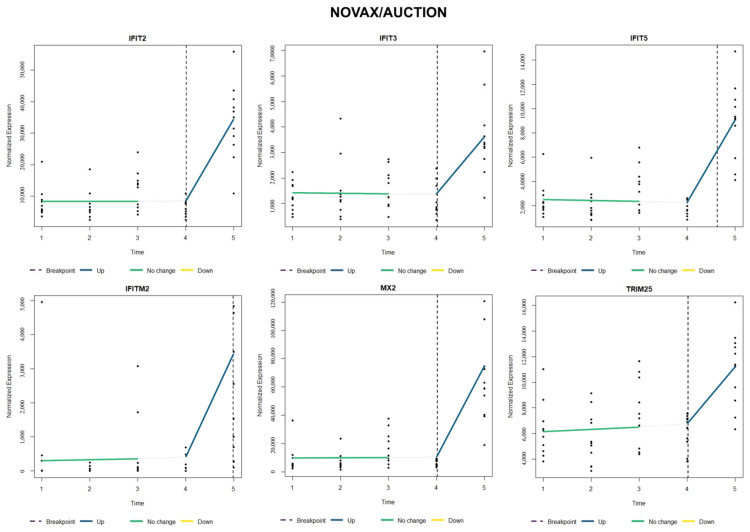
Time-course expression of type I interferon-stimulated genes (ISGs) following NOVAX/AUCTION treatment. Normalized expression levels of selected ISGs (*IFIT2*, *IFIT3*, *IFIT5*, *IFITM2*, *MX2*, and *TRIM25*) are shown across all five time points (T1–T5). Each panel represents one gene, with relative sample expression values plotted as black dots along the y-axis. Colored lines indicate estimated trends over time: blue for significant upregulation, yellow for significant downregulation, green for no significant change, and white for the estimated trend within that segment prior to the breakpoint. Vertical dashed lines represent the estimated breakpoint, suggesting a critical shift in gene expression levels.

**Figure 4 vetsci-12-00834-f004:**
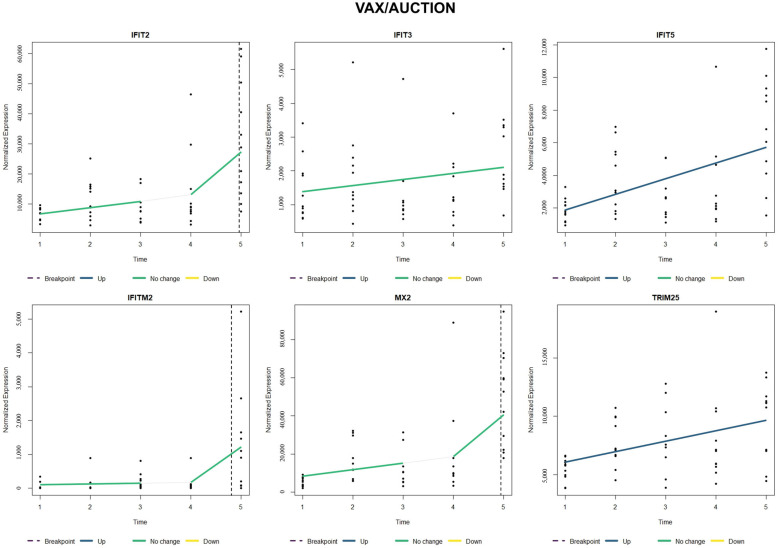
Time-course expression of type I interferon-stimulated genes (ISGs) following VAX/AUCTION treatment. Normalized expression levels of selected ISGs (*IFIT2*, *IFIT3*, *IFIT5*, *IFITM2*, *MX2*, and *TRIM25*) are shown across all five time points (T1–T5). Each panel represents one gene, with relative sample expression values plotted as black dots along the y-axis. Colored lines indicate estimated trends over time: blue for significant upregulation, yellow for significant downregulation, green for no significant change, and white for the estimated trend within that segment prior to the breakpoint. Vertical dashed lines represent the estimated breakpoint, suggesting a critical shift in gene expression levels.

**Figure 5 vetsci-12-00834-f005:**
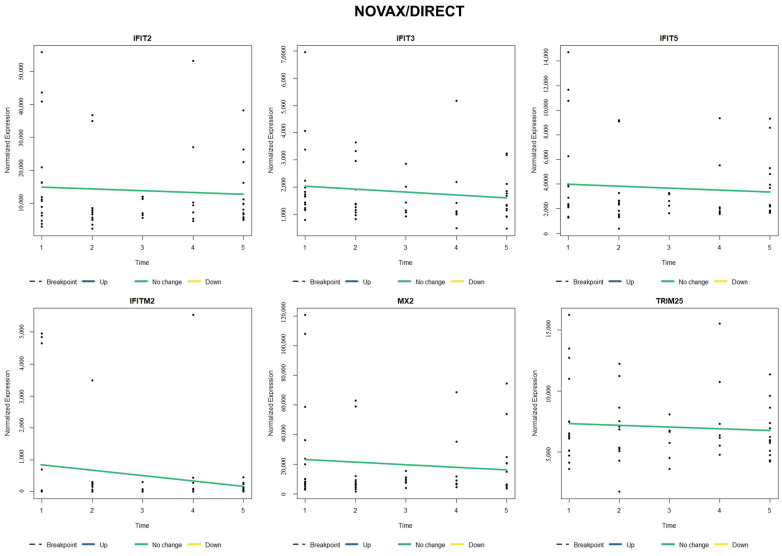
Time-course expression of type I interferon-stimulated genes (ISGs) following NOVAX/DIRECT treatment. Normalized expression levels of selected ISGs (*IFIT2*, *IFIT3*, *IFIT5*, *IFITM2*, *MX2*, and *TRIM25*) are shown across all five time points (T1–T5). Each panel represents one gene, with relative sample expression values plotted as black dots along the y-axis. Colored lines indicate estimated trends over time: blue for significant upregulation, yellow for significant downregulation, green for no significant change, and white for the estimated trend within that segment prior to the breakpoint. Vertical dashed lines represent the estimated breakpoint, suggesting a critical shift in gene expression levels.

**Figure 6 vetsci-12-00834-f006:**
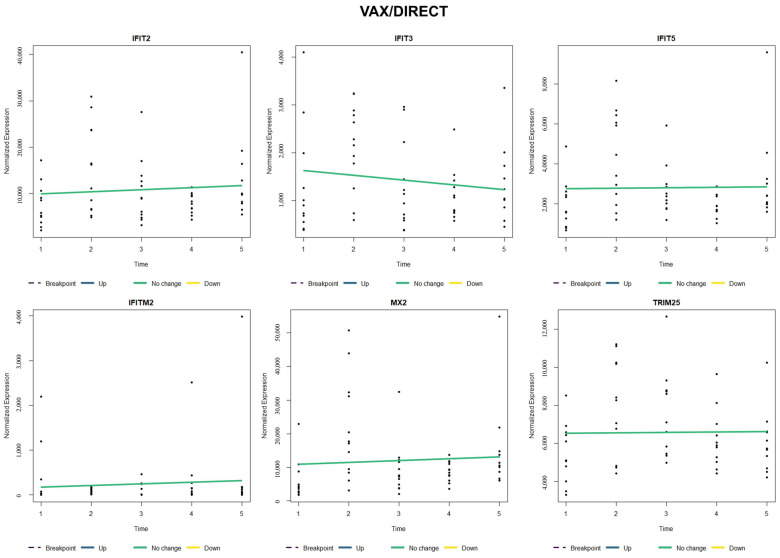
Time-course expression of type I interferon-stimulated genes (ISGs) following VAX/DIRECT treatment. Normalized expression levels of selected ISGs (*IFIT2*, *IFIT3*, *IFIT5*, *IFITM2*, *MX2*, and *TRIM25*) are shown across all five time points (T1–T5). Each panel represents one gene, with relative sample expression values plotted as black dots along the y-axis. Colored lines indicate estimated trends over time: blue for significant upregulation, yellow for significant downregulation, green for no significant change, and white for the estimated trend within that segment prior to the breakpoint. Vertical dashed lines represent the estimated breakpoint, suggesting a critical shift in gene expression levels.

**Figure 7 vetsci-12-00834-f007:**
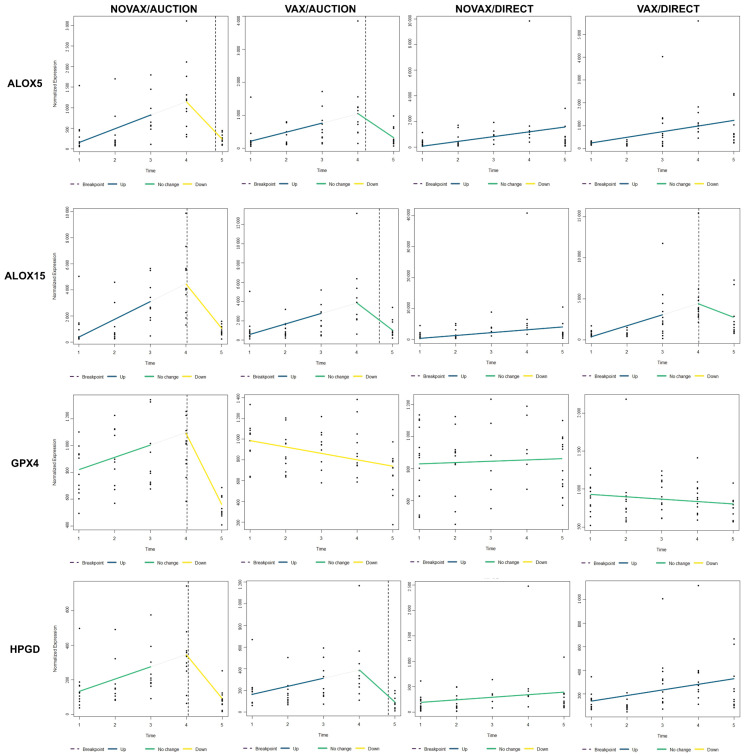
Time-course expression of specialized pro-resolving mediator (SPM)-related genes across all treatment conditions. Normalized expression levels of *ALOX5*, *ALOX15*, *GPX4*, and *HPGD* are displayed across four treatment conditions: NOVAX/AUCTION, VAX/AUCTION, NOVAX/DIRECT, and VAX/DIRECT. Each panel represents one gene-treatment combination, with expression measured over all five time points (T1-T5). Individual sample values are shown as black dots along the y-axis. Trend lines indicate directional changes: blue for significant upregulation, yellow for significant downregulation, green for no significant change, and white for the estimated trend within that segment prior to the breakpoint. Vertical dashed lines represent the estimated breakpoint, suggesting a critical shift in gene expression levels.

## Data Availability

The data presented in the study are deposited in the National Center for Biotechnology Information Gene Expression Omnibus (NCBI-GEO), accession number GSE248477. Previously generated data utilized by this study are found in the NCBI-GEO, under accession numbers GSE205004 and GSE218061. All other relevant data are within the paper and its Appendix A.

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
