# Peer review of "Proinflammatory Cytokines, Type I Interferons, and Specialized Proresolving Mediators Hallmark the Influence of Vaccination and Marketing on Backgrounded Beef Cattle"

_vetsci, 2025, doi:10.3390/vetsci12090834_

Round 1

Reviewer 1 Report

Comments and Suggestions for Authors

The authors demonstrated how vaccination, marketing strategy, and temporal gene expression in young beef cattle are important e strictly correlated each other key factors. It is a very good structured and deeply analyzed survey, with a strong dataset and provides good practical management operations.

The presentation of results and goal are clear regardless the languages seems to be pretty confused and not fluid.   I really appreciate the study design and the discussion, the topic is pertinent and the reference are suitable.   It is definitely a not ordinary research, but I appreciate

Author Response

Thank you for taking the time and reviewing our manuscript.

Reviewer 2 Report

Comments and Suggestions for Authors

In present work, Hudson et al. try to study the proinflammatory cytokine production, type I interferon signaling, and specialized proresolving mediator synthesis that hallmarks the influence of attenuated viral vaccination and marketing strategies on backgrounded beef cattle. This study provides mechanistic insight into how routine management practices influence immunological resilience and highlights the value of integrating transcriptomics into bovine respiratory disease risk mitigation strategies within beef production systems. However, there are some questions that should be explained.

Major concerns

  1. Please explain why five time points (107 days, 183 days, 114 days, 230 days, and four days post-weaning) are used in this study.
  2. In this study, differential expression and gene trajectory analyses were performed. However, the results should be further confirmed using PCR for key genes or western blot for key proteins.
  3. The detail for total RNA isolated from whole blood should be added.
  4. English grammar and writing should be checked and revised throughout the manuscript. There are so many long sentences that are difficult to understand.

Minor concerns

  1. Line 1, change ‘Type of the Paper (Article)’ to ‘Article’.
  2. The title is too long, which is difficult to understand and should be refined.
  3. Simple Summary section should be rewritten. There is no a background. Please explain ‘BRD’ in line 32.
  4. Abstract section is too long, which should be refined.
  5. Keywords should be revised. Too many Keywords.
  6. A hypothesis should be added in the end of Introduction section.
  7. Materials and Methods section, number the subsections.
  8. Lines 105-107 and 577-579, it is repeated.
  9. Lines 110-157 and 199-226, too long paragraphs.
  10. A Statistical analyses subsection should be added in the Materials and Methods section.
  11. Figures 1-7 are too large, and not completely displayed.
  12. A conclusion section is needed.
  13. The reference format is not consistent. Journal name of some references are in abbreviation, but some references are not. Please check these throughout Reference section. Reference 11, ‘ajvr’ is not suitable.
Comments on the Quality of English Language

The English could be improved to more clearly express the research.

Author Response

Open Review

Quality of English Language

(x) The English could be improved to more clearly express the research.
( ) The English is fine and does not require any improvement.

Yes

Can be improved

Must be improved

Not applicable

Does the introduction provide sufficient background and include all relevant references?

( )

( )

(x)

( )

Is the research design appropriate?

( )

( )

(x)

( )

Are the methods adequately described?

( )

( )

(x)

( )

Are the results clearly presented?

( )

( )

(x)

( )

Are the conclusions supported by the results?

( )

( )

(x)

( )

Are all figures and tables clear and well-presented?

( )

( )

(x)

( )

Comments and Suggestions for Authors

In present work, Hudson et al. try to study the proinflammatory cytokine production, type I interferon signaling, and specialized proresolving mediator synthesis that hallmarks the influence of attenuated viral vaccination and marketing strategies on backgrounded beef cattle. This study provides mechanistic insight into how routine management practices influence immunological resilience and highlights the value of integrating transcriptomics into bovine respiratory disease risk mitigation strategies within beef production systems. However, there are some questions that should be explained.

Major concerns

  1. Please explain why five time points (107 days, 183 days, 114 days, 230 days, and four days post-weaning) are used in this study.

Please see lines 161-163 for additional clarification added based on this comment. Five timepoints were selected to mirror common beef industry management practices of initial respiratory vaccination or branding vaccination, revaccination, and weaning, while also capturing key biological windows for immune development. Maternal antibody protection typically declines between 2-6 months age therefore time point 1 (approx. 3.5 months) was chosen to reflect that most calves, regardless of individual variation in antibody decline, would benefit from the administration of an attenuated viral vaccination (Chase et al., 2008). Timepoint 2 (7 days later) intended to measure the acute immune response to this initial vaccination. Timepoint 3 corresponds to revaccination, which is recommended to occur prior to potential pathogen exposure, allowing adequate time (approximately 3 weeks) for an adaptive immune response to develop (Richeson and Falkner, 2020). Timepoints 4 and 5 were designed to capture post-vaccination and marketing-related effects on gene expression, health, and performance. This structure mirrors a parallel component of the larger project that evaluated inflammatory mediator profiles alongside health and performance outcomes at the same intervals over several replicates (i.e., years; manuscript forthcoming).

Chase, C. C. L., Hurley, D. J., & Reber, A. J. (2008). Neonatal immune development in the calf and its impact on vaccine response. Veterinary Clinics of North America: Food Animal Practice, 24(1), 87–104.

Richeson JT, Falkner TR. Bovine Respiratory Disease Vaccination: What Is the Effect of Timing? Vet Clin North Am Food Anim Pract. 2020 Jul;36(2):473-485. doi: 10.1016/j.cvfa.2020.03.013. PMID: 32451036.

  1. In this study, differential expression and gene trajectory analyses were performed. However, the results should be further confirmed using PCR for key genes or western blot for key proteins.

The authors agree that qPCR and Western blotting are valuable tools for validating molecular findings in follow-up studies, particularly for hypothesis-driven investigations of individual pathways (Zhide and Cui, 2011). However, the primary objective of this study was to characterize broad gene expression patterns and pathway-level responses associated with vaccination and marketing strategy, rather than quantify specific transcript or protein abundance. RNA-sequencing provided an unbiased, genome wide profile capable of detecting differential expression across thousands of genes simultaneously with higher sensitivity and dynamic range than qPCR (Wang et al., 2009). Additionally, it is also important to note that mRNA and protein abundance are not always directly correlated, as post-transcriptional and translational regulation, as well as protein stability, can influence final protein levels (Liu et al., 2016). While PCR and Western blotting provide targeted validation, they measure related but distinct areas of regulation that are not always directly comparable to RNA-Seq derived expression. Further, qPCR is inherently limited to a predefined subset of genes, whereas our transcriptomic approach was designed and appropriately powered to uncover systemic patterns, immune pathways, and temporal dynamics. Finally, the repeated measures structure of five timepoints would render targeted qPCR or western blotting impractical within the scope and resources of this project. We agree that future research may build upon these findings by employing qPCR or Western blotting in targeted, hypothesis-driven studies to validate specific pathways or mechanisms across larger and more diverse populations (i.e., dairy cattle, breeding facility animals, etc.).

Zhide Fang, Xiangqin Cui, Design and validation issues in RNA-seq experiments, Briefings in Bioinformatics, Volume 12, Issue 3, May 2011, Pages 280–287, https://doi.org/10.1093/bib/bbr004

Wang Z, Gerstein M, Snyder M. RNA-Seq: a revolutionary tool for transcriptomics. Nat Rev Genet. 2009 Jan;10(1):57-63. doi: 10.1038/nrg2484.

Liu, Y., Beyer, A., & Aebersold, R. 2016. On the Dependency of Cellular Protein Levels on mRNA Abundance. Cell, 165. doi:10.1016/j.cell.2016.03.014

  1. The detail for total RNA isolated from whole blood should be added.

Detail for RNA extractions can be found on L175-179 including concentrations and RNA integrity number (RIN) which is a scaled value that is used to assess the quality and integrity of the samples.

  1. English grammar and writing should be checked and revised throughout the manuscript. There are so many long sentences that are difficult to understand.

Revisions have been made throughout the manuscript.

Minor concerns

  1. Line 1, change ‘Type of the Paper (Article)’ to ‘Article’. Corrections were made.
  2. The title is too long, which is difficult to understand and should be refined. Corrections were made to improve the conciseness of the article title.
  3. Simple Summary section should be rewritten. There is no a background. Please explain ‘BRD’ in line 32. Corrections were made to include background information regarding relevance to the approach/findings and to define BRD.
  4. Abstract section is too long, which should be refined. Corrections were made to improve the conciseness of the abstract section.
  5. Keywords should be revised. Too many Keywords. Corrections were made to emphasize keywords.
  6. A hypothesis should be added in the end of Introduction section. Additional information was added to clarify our hypothesis on L107-109.
  7. Materials and Methods section, number the subsections. All subsections have been numbered.
  8. Lines 105-107 and 577-579, it is repeated.

The animal use statement was removed from the body of work.

  1. Lines 110-157 and 199-226, too long paragraphs.

Corrections have been made.

  1. A Statistical analyses subsection should be added in the Materials and Methods section.

We appreciate the reviewer’s suggestion regarding the statistical analyses. However in this study, statistical analysis were integrated directly within the RNA-sequencing bioinformatics pipeline rather than presented in a stand-alone approach and section. Specifically, gene-level count matrices were processed with edge R v3.36.0 under a generalized linear model (GLM) framework, with dispersion estimated and quasi-likelihood F tests applied to test for differential expression between treatment groups and time points. For functional enrichment analyses, differentially expressed genes were tested against bovine-specific GO, KEGG, and Reactome databases using KOBAS-i, specifically through hypergeometric testing with Benjamini-Hochberg correction (FDR <0.05). To capture dynamic temporal expression patterns, we additionally applied the Trendy v1.26.0 segmented regression model, which evaluated gene trajectories across time-course high throughput data. Importantly, each modeling workflow (i.e., differential gene expression analysis, functional enrichment analysis, etc.) models with differing distributions and error-rate assumptions. The manner in which we elected to describe each workflow enhances the transparency in our approach and allows for increased detail for possible reproduction of results. While not presented as a separate statistics section, statistical testing was integrated at each step of the transcriptomic analysis pipeline, consistent with current best practices in high dimensional -omics research.

  1. Figures 1-7 are too large, and not completely displayed.

We believe this to have been a typesetting error from the provided template and document conversion. Corrections have been made.

  1. A conclusion section is needed.

The authors appreciate the reviewer’s suggestion regarding a conclusion section. The journal guidelines do not require a standalone conclusion section, and, given the structure of our manuscript, we feel that key interpretations and implications were already clearly integrated into the discussion and study limitations sections. We strove to avoid redundancy and maintain conciseness; therefore, we did not include a separate conclusion section.

  1. The reference format is not consistent. Journal name of some references are in abbreviation, but some references are not. Please check these throughout Reference section. Reference 11, ‘ajvr’ is not suitable.

We apologize for these typographical errors. Corrections have been made throughout the reference section.

Reviewer 3 Report

Comments and Suggestions for Authors

This manuscript presents a robust and relevant investigation into how preweaning viral vaccination and marketing strategies (direct vs. auction) influence the immunological gene expression landscape in beef calves. The authors use a split-plot randomized controlled design and RNA-seq-based transcriptomics to analyze longitudinal immune responses in clinically healthy animals. The integration of pathway enrichment, time-course analysis (via Trendy), and differentially expressed gene profiles yields important insights into BRD resilience markers and the stress associated with marketing routes. However, some concerns related to figure clarity, group size balance, overinterpretation, and insufficient pathogen data need to be addressed before acceptance.

  1. The abstract is too long (nearly 30 lines) and dense. Condense to highlight only core methods, major findings, and implications.
  2. Please number your subsections.
  3. Please clearly describe exclusion criteria and whether subclinical infections (e.g., via serology or viral PCR) were ruled out.
  4. Authors frequently hypothesize viral exposure in the AUCTION group but did not measure it. Acknowledge this as a limitation more directly. Ideally, suggest incorporating pathogen detection (e.g., PCR or metagenomics) in future studies to confirm exposure.
  5. The manuscript interprets gene expression stability in VAX/AUCTION as "buffering immune dysregulation", but correlation does not equal causation. Soften language throughout to reflect this is a correlation, not direct proof of protective modulation (e.g., use “suggests” or “may indicate”).
  6. Only male calves were included, which limits generalizability. Justify why only males were used (castration-related standardization?) and include this clearly under limitations.
  7. All figures included in the manuscript (Figures 1 through 7) appear to be incomplete or cropped, with substantial portions of the images missing. This includes missing axes, gene labels, legends, trend lines, and in some cases, entire panels or time points. For example, the UpSet plot in Figure 1 lacks visible axis labels and complete set intersections, while Figures 3 to 7, which present time-course expression data, are truncated, making it impossible to interpret gene trajectories or assess statistical trends. The heatmap in Figure 2 is also cut off, obscuring cluster annotations and sample groupings. These visualizations are essential for validating the transcriptomic findings, and their current incomplete state significantly limits the reader’s ability to evaluate the data. The authors are strongly encouraged to reinsert all figures at full resolution, ensuring that all graphical elements are intact and clearly labeled. Figures should also be submitted as high-quality standalone files to avoid layout distortion during typesetting.

Author Response

Open Review

Quality of English Language

( ) The English could be improved to more clearly express the research.
(x) The English is fine and does not require any improvement.

Yes

Can be improved

Must be improved

Not applicable

Does the introduction provide sufficient background and include all relevant references?

(x)

( )

( )

( )

Is the research design appropriate?

( )

(x)

( )

( )

Are the methods adequately described?

( )

(x)

( )

( )

Are the results clearly presented?

( )

(x)

( )

( )

Are the conclusions supported by the results?

(x)

( )

( )

( )

Are all figures and tables clear and well-presented?

( )

(x)

( )

( )

Comments and Suggestions for Authors

This manuscript presents a robust and relevant investigation into how preweaning viral vaccination and marketing strategies (direct vs. auction) influence the immunological gene expression landscape in beef calves. The authors use a split-plot randomized controlled design and RNA-seq-based transcriptomics to analyze longitudinal immune responses in clinically healthy animals. The integration of pathway enrichment, time-course analysis (via Trendy), and differentially expressed gene profiles yields important insights into BRD resilience markers and the stress associated with marketing routes. However, some concerns related to figure clarity, group size balance, overinterpretation, and insufficient pathogen data need to be addressed before acceptance.

  1. The abstract is too long (nearly 30 lines) and dense. Condense to highlight only core methods, major findings, and implications. Similar to suggestions made by Reviewer 2, we have made corrections to the abstract to improve consciousness and readability.
  2. Please number your subsections. Corrections were made; all subsections are numbered.
  3. Please clearly describe exclusion criteria and whether subclinical infections (e.g., via serology or viral PCR) were ruled out. Additional clarification was added on L158-160. While the visual assessment of clinical illness is the most common practice in production systems, further research utilizing and emphasizing more objective antemortem metrics of disease (e.g., transthoracic ultrasonography, airway cytology, etc.) is highly warranted. We briefly discuss our limitations regarding this point in L547-555.
  4. Authors frequently hypothesize viral exposure in the AUCTION group but did not measure it. Acknowledge this as a limitation more directly. Ideally, suggest incorporating pathogen detection (e.g., PCR or metagenomics) in future studies to confirm exposure.  Thank you for this comment. We directly addressed this topic on L507-509 and have added L553-555.
  5. The manuscript interprets gene expression stability in VAX/AUCTION as "buffering immune dysregulation", but correlation does not equal causation. Soften language throughout to reflect this is a correlation, not direct proof of protective modulation (e.g., use “suggests” or “may indicate”).  We appreciate the reviewer’s suggestion regarding causal components to our work. Language has been edited throughout the manuscript.
  6. Only male calves were included, which limits generalizability. Justify why only males were used (castration-related standardization?) and include this clearly under limitations.  

The authors acknowledge that the inclusion of only male calves limits generalizability of these findings to mixed sex populations. However, male calves were selected for this study to minimize variability in management practices and closely reflect industry demographics, where steers comprise more than half of all beef breed or crossbred cattle placed in U.S feedlots with 1000 head or more (USDA-APHIS, 2011). Additionally, the broader trial from which these animals were sampled incorporated castration as part of the management protocol at time of revaccination (T3). The incorporation of only male calves ensures uniformity in castration status, while also controlling for potential sex and procedure related effects to stress physiology, growth performance and immune gene expression (Roberts et al., 2018, Pang et al., 2009). Importantly, our a priori power calculation indicated that the study was adequately powered to detect differential expression despite this restriction. We have now clarified this rationale and added the limitation regarding sex-specific generalizability to the manuscript (L545-547).

USDA. Feedlot 2011 “Part I: Management Practices on U.S. Feedlots with a Capacity of 1,000 or More Head.” USDA–APHIS–VS–CEAH–NAHMS. Fort Collins, CO #626.0313

Roberts SL, Powell JG, Hughes HD, Richeson JT. Effect of castration method and analgesia on inflammation, behavior, growth performance, and carcass traits in feedlot cattle. J Anim Sci. 2018 Feb 15;96(1):66-75. doi: 10.1093/jas/skx022.

Pang W, Earley B, Sweeney T, Gath V, Crowe MA. Temporal patterns of inflammatory gene expression in local tissues after banding or burdizzo castration in cattle. BMC Vet Res. 2009 Sep 23;5:36. doi: 10.1186/1746-6148-5-36.

  1. All figures included in the manuscript (Figures 1 through 7) appear to be incomplete or cropped, with substantial portions of the images missing. This includes missing axes, gene labels, legends, trend lines, and in some cases, entire panels or time points. For example, the UpSet plot in Figure 1 lacks visible axis labels and complete set intersections, while Figures 3 to 7, which present time-course expression data, are truncated, making it impossible to interpret gene trajectories or assess statistical trends. The heatmap in Figure 2 is also cut off, obscuring cluster annotations and sample groupings. These visualizations are essential for validating the transcriptomic findings, and their current incomplete state significantly limits the reader’s ability to evaluate the data. The authors are strongly encouraged to reinsert all figures at full resolution, ensuring that all graphical elements are intact and clearly labeled. Figures should also be submitted as high-quality standalone files to avoid layout distortion during typesetting

We appreciate the reviewer’s comments regarding manuscript figures. Similar to Reviewer 2’s comments, we believe this to be a typesetting error upon file conversion and have corrected the images to be fully included into the manuscript body.

Round 2

Reviewer 2 Report

Comments and Suggestions for Authors

Thanks for author’s responses. However, English grammar and writing STILL should be checked and revised throughout the manuscript.

For example,

Line 45, correct ‘HISAT2/StringTie2 .’.

Line 49, correct ‘MX2, TRIM25’ to ‘MX2, and TRIM25’.

Line 49, correct ‘ALOX5, ALOX15’ to ‘‘ALOX5 and ALOX15’’.

Line 194, delete ‘(TMM)’.

Line 197, delete ‘(GLM)’.

Lines 205 and 281, there are two ‘gene ontology (GO)’.

All Figures, figure titles and figure legends are not obvious.

Line 577, delete ‘[14,15].’.

References section, the abbreviations of some journal names are not right. For example,

Ref. 2, correct ‘Vet Microbio’ to ‘Vet Microbiol’.

Ref. 6, correct ‘J of Dairy Sci’ to ‘J Dairy Sci’.

Ref. 10, 34, 42, and 45, correct ‘J of Anim Sci’ to ‘J Anim Sci’.

Ref. 11, correct ‘Amer J of Vet Res’ to ‘Am J Vet Res’.

Ref. 52, correct ‘Jl of Expl Med’ to ‘J Exp Med’.

Please check these throughout Reference section one by one.

Comments on the Quality of English Language

The English could be improved to more clearly express the research.

Author Response

(x) I would not like to sign my review report
( ) I would like to sign my review report Quality of English Language (x) The English could be improved to more clearly express the research.
( ) The English is fine and does not require any improvement.            
  Yes Can be improved Must be improved Not applicable
Does the introduction provide sufficient background and include all relevant references? ( ) (x) ( ) ( )
Is the research design appropriate? ( ) (x) ( ) ( )
Are the methods adequately described? ( ) (x) ( ) ( )
Are the results clearly presented? ( ) (x) ( ) ( )
Are the conclusions supported by the results? ( ) (x) ( ) ( )
Are all figures and tables clear and well-presented? ( ) (x) ( ) ( )
    Comments and Suggestions for Authors

Thanks for author’s responses. However, English grammar and writing STILL should be checked and revised throughout the manuscript.

For example,

Line 45, correct ‘HISAT2/StringTie2 .’.

Line 49, correct ‘MX2, TRIM25’ to ‘MX2, and TRIM25’.

Line 49, correct ‘ALOX5, ALOX15’ to ‘‘ALOX5 and ALOX15’’.

Line 194, delete ‘(TMM)’.

Line 197, delete ‘(GLM)’.

Lines 205 and 281, there are two ‘gene ontology (GO)’.

All Figures, figure titles and figure legends are not obvious.

Line 577, delete ‘[14,15].’.

References section, the abbreviations of some journal names are not right. For example,

Ref. 2, correct ‘Vet Microbio’ to ‘Vet Microbiol’.

Ref. 6, correct ‘J of Dairy Sci’ to ‘J Dairy Sci’.

Ref. 10, 34, 42, and 45, correct ‘J of Anim Sci’ to ‘J Anim Sci’.

Ref. 11, correct ‘Amer J of Vet Res’ to ‘Am J Vet Res’.

Ref. 52, correct ‘Jl of Expl Med’ to ‘J Exp Med’.

Please check these throughout Reference section one by one.

The authors appreciate your comments and have reevaluated language throughout the manuscript. All suggested corrections have been made.

Reviewer 3 Report

Comments and Suggestions for Authors

The quality of the current manuscript has been improved greatly. I now agree for further process. Congratulations!

Author Response

    Open Review ( ) I would not like to sign my review report
(x) I would like to sign my review report Quality of English Language ( ) The English could be improved to more clearly express the research.
(x) The English is fine and does not require any improvement.            
  Yes Can be improved Must be improved Not applicable
Does the introduction provide sufficient background and include all relevant references? (x) ( ) ( ) ( )
Is the research design appropriate? (x) ( ) ( ) ( )
Are the methods adequately described? (x) ( ) ( ) ( )
Are the results clearly presented? (x) ( ) ( ) ( )
Are the conclusions supported by the results? (x) ( ) ( ) ( )
Are all figures and tables clear and well-presented? (x) ( ) ( ) ( )
    Comments and Suggestions for Authors

The quality of the current manuscript has been improved greatly. I now agree for further process. Congratulations!

Thank you very much and the authors look forward to publication.